# RADAR AND PFIM: CONTENT-PRIOR PATCH-SPACE MODULATION FOR EFFICIENT VISION TRANSFORMERS

## ABSTRACT

Positional encodings in Vision Transformers, relative (iRPE, ROPE) or otherwise, help to reason about space but remain content-agnostic. We introduce a lightweight, content-aware patch modulation that injects a quasi-positional prior computed from pre-trained patch embeddings. We present two light weight, drop-in pre-MHSA modules: **RADAR** (anchor-conditioned distance priors that modulate tokens) and **PFIM** (parameter-free importance scaling with **no new trainable** parameters beyond the logit layer). Both keep the ViT backbone frozen, preserve the **attention kernel**, and add negligible to no overhead.

On CIFAR-100 with absolute positional encoding, RADAR boosts Top-1 accuracy by **+7.5 pp** and Top-5 by **+3.3 pp** over vanilla ViT, and by **+4.1 pp / +1.6 pp** over a strong single-CPE baseline. PFIM improves vanilla ViT by **+2.0 pp** (Top-1) and **+1.1 pp** (Top-5), performing on par with Single-PEG within a small margin. Improvements are statistically significant across seeds (paired t-test, 95% CI). RADAR contains **56%** and PFIM **88%**, **fewer** trainable params compared to Single-PEG on CIFAR100. By turning latent patch geometry into content-aware priors, our approach reallocates attention to semantically relevant regions, offering parameter-efficient gains ideal for low-budget training. Code for ablations & experiments will be shared.

## 1 INTRODUCTION

Vision Transformers (ViTs) Dosovitskiy et al. (2020) use self-attention to capture long-range dependencies but, being permutation-invariant, require explicit positional signals for robustness.

While order mechanisms such as "Absolute Positional Embeddings" were introduced in Dosovitskiy et al. (2020) ; they're considered less robust to methods that add "Relative Positional Embeddings" e.g., Conditional Position Encoding (CPE) Chu & Tian (2021) and iRPE Peng et al. (2021). But almost all (relative/absolute) methods requires MHSA ( Mulit-Headed Self Attention ) to learn attention mapping on tokens from scratch and largely lack content-aware global structure.

We address this by inducing **patch-space global guidance**. We compute lightweight priors directly from pre-trained patch embeddings and inject them pre-MHSA : (i) anchor-conditioned relative distance priors (RADAR), and (ii) parameter-free importance scaling (PFIM). These global, content-aligned cues shape attention before the first $QK^T$, helping them in starting informed, improving accuracy and stability under tight compute and memory budgets while leaving the ViT backbone unchanged.

We introduce two complementary drop-in methods:

1. **RADAR** - Relational Anchor-Distance Attentional Re-weighting: We compute compact, content-relevant global embeddings and **modulate** patch embeddings with them. MHSA no longer has to infer long-range context from scratch; it is nudged towards salient structure from the outset. RADAR is architecture-agnostic, adds only minimal overhead, and consistently outperforms strong ViT and local PE baselines.

2. **PFIM** - Parameter-Free Importance Modulation A simple yet potent scheme that scales patch embeddings by data-driven importance -**with zero new trainable parameters beyond the final**

**logit layer**. PFIM exploits intrinsic patch statistics to amplify informative regions and dampen distractors, delivering robust gains in low-resource and compute settings, outperforming Vanilla ViT and being on-par with Single-PEG.

RADAR has two variants: (i) **Single Soft Anchor** (probability-weighted sum) distance injection, and (ii) **Leave-one-out Soft Anchor** that excludes token $j$ when computing the weighted sum of remaining $j - 1 to N$; in both methods, we compute polynomial features of their distances with anchor and **FiLM**-modulate tokens before the encoder/self-attention: $x'_j = (1 + \alpha * s_j) \odot x_j + \beta \odot b_j$.

In PFIM, importance scores (based on entropy of patch embeddings) per sequence scale embeddings via sign-aware powers and are residually mixed with originals: $x_j = (1 - mix\%) * x_j + mix\% * x_j^{pow}$.

Our contributions are as follows:

- **Parameter-efficient attention guidance**. We show that injecting patch-space priors **before** encoder/MHSA yields better accuracy than learning everything end-to-end.

- **Two practical, budget-friendly modules**. (i) **RADAR:** global, relational Anchor-Distance content priors, for token modulation with negligible extra parameters and runtime cost; (ii) **PFIM:** a parameter-free importance scaler requiring no architectural changes and only the standard logit layer.

- **Backbone-agnostic, drop-in design**. Both methods preserve the ViT encoder, integrate with a single pre-MHSA step, and are well-suited to constrained training regimes.

Together, RADAR and PFIM show that strategic pre-MHSA guidance strengthens ViTs without larger models, deeper stacks, or costly training. Injecting pre-encoder/MHSA is also train-time robust, tolerating smoothing of up to **50%** of token-importance weights, aided by downstream encoder layer norms.

## 2 RELATED WORK

### 2.1 TRANSFORMERS FOR VISION AND POSITIONAL INFORMATION

ViT showed that patch tokens with global self-attention can work well for images, which made positional signals a central design choice in vision Transformers (Dosovitskiy et al., 2021). DeiT made this approach data-efficient on ImageNet and introduced a distillation token (Touvron et al., 2021a). A growing body of work designs position signals that enter attention directly rather than through token addition. This includes relative position representations, T5's bucketed relative bias, ALiBi's distance-linear bias for length extrapolation, RoPE's rotary embedding, and more which have become popular in recent years (Shaw et al., 2018; Raffel et al., 2020; Press et al., 2021; Su et al., 2021). In vision, Swin adds a 2D relative bias inside shifted windows and Swin V2 introduces a log-spaced continuous bias for resolution transfer (Liu et al., 2021; 2022). A key issue is that most of these position designs are fixed with respect to content or stay local. They bias attention by distance or windows but do not tell the model which regions are salient in the current image.

### 2.2 CONTENT-CONDITIONED LOCALITY AND LIGHTWEIGHT BIAS

ConViT initializes attention with a soft convolutional prior that can relax towards global attention (d'Ascoli et al., 2021). LocalViT injects depthwise convolutions into the MLP to add locality (Li et al., 2021). CPVT conditions positional encodings on nearby tokens with a tiny PEG (Chu et al., 2023). These works improve sample efficiency and stability by guiding where to look.

CPVT and LocalViT inject local cues into patch tokens, reducing reliance on static positional encodings, but their $3x3$ PEGs remain coarse and miss global context. RADAR supplies content-aware global priors that guide attention across the whole image. For an apples-to-apples comparison under APE (required by our method), we adopt CPVT's recommended Single-PEG at the 0th layer with a $27x27$ convolution—expanding the receptive field while retaining APE. This forms a strong local baseline yet still lacks RADAR's global context, underscoring our method's robustness.

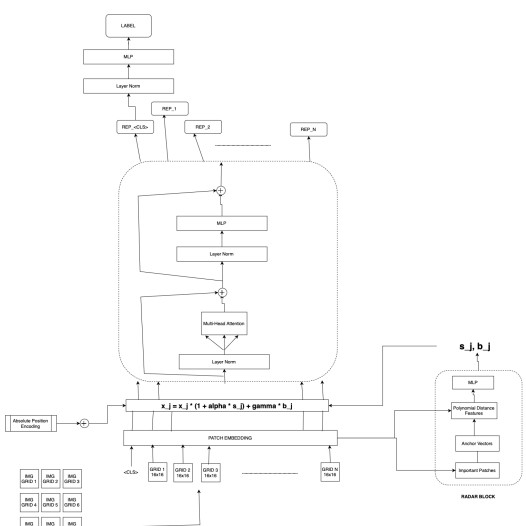

Figure 1: RADAR Architecture

### 2.3 ANCHORS, LANDMARKS, AND LATENT BOTTLENECKS

Some models insert a small set of learned anchor or latent tokens that sit between all tokens and pass information back and forth. Set Transformer uses inducing points for permutation-invariant sets (Lee et al., 2019). Perceiver alternates cross-attention through a fixed latent bottlenceck (Jaegle et al., 2021). Nyströmformer uses landmark points to approximate attention in linear time (Xiong et al., 2021). CaiT improves deep ViTs with class-attention layers that help the class token aggregate information (Touvron et al., 2021b). Other recent work routes or sparsifies attention, including BiFormer and Star-Transformer (Zhu et al., 2023; Guo et al., 2019). These methods restructure or approximate attention for efficiency. They typically change the attention rule or add routing. RADAR keeps the backbone intact and uses anchors only to compute relational distances that guide the existing attention.

RADAR, while inspired by FiLM, is distinct: in FiLM the $(s_j, b_j)$ vectors encode task-dependent features for CNN activations, whereas in RADAR they are derived from relative distance features in patch space. Moreover, FiLM was designed for CNNs, while RADAR adapts this modulation principle to ViTs with global, content-aware anchors (Perez et al., 2018)..

### 2.4 PARAMETER-EFFICIENT ADAPTATION

Researchers have explored adapting vision Transformers cheaply by adding minimal new parameters. Their adapters add tiny bottlenecks between layers and train only those parameters. LoRA injects low-rank updates into attention or MLP weights while freezing the backbone . Visual Prompt Tuning learns a few prompt tokens. AdaptFormer tailors lightweight adapters for ViTs and videos . These methods reduce per-task cost while preserving strong pretrained models.

## 3 METHODOLOGY

### 3.1 ANCHOR DISTANCE METHODS

We explored ways like: Arg max of sequences, arg min, norm - Soft max, Entropy etc to identify important regions of patch embedding tokens. Anchor Methods, use this 'important region' and aggregate patch embeddings with these scores, which are then used to calculate distances with patch embeddings and subsequently inject these distances into them in a specific style.

---

**Algorithm 1** RADAR: Anchor Distance Method

---

1: Given $X \in \mathbb{N}xD$, compute aggregate vectors, $vector\_values \in \mathbb{N}$, containing raw score per token.
2: Compute Anchor Vectors:
      a) Single Soft Anchor: $\sum(X_j * vector\_values_j)$ $j \in \{1, \ldots, N\}$ **(OR)**
      b) Leave-One-Out Soft Anchor: $\sum(X_j * vector\_values_j)j \notin i$.
3: $Distance[Patch\_Embeddings, Anchor\_Vectors]$. $offset=\sqrt{(X_j - anchors)^2 + delta^2} - delta$.
4: Polynomial Feature Bank: $\phi = [sin(offset), \cos(offset), \log(offset), offset^2, \sqrt{(offset)}]$
5: Projected Scale and Shift: $s_j, b_j = MLP(\phi) \in (D)$
6: FiLM-Modulate Patch Embeddings: $x'_j = (1 + \alpha * s_j) \odot x_j + \beta \odot b_j$.

---

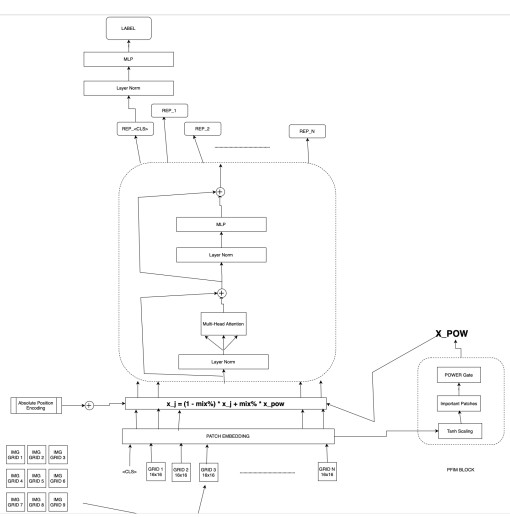

Figure 2: PFIM Architecture

## 3.2 WHY ANCHOR DISTANCES

1. $vector\_values$ produces scalar importance weights per token. These are unstructured scores—they rank salience but don't yield a representation that can interact geometrically with other tokens.

2. $SSA$ or $LOOSA$ alone cant' represent global , relative information. Anchor vectors (importance-weighted patch embeddings); exist in the same feature space as the patches, but without $offset$ and $\phi$ they remain raw embeddings. They don't yet encode relational geometry—the pairwise or anchored distances that are essential for providing global context of patch space; just "contextualized tokens," not distance features.

3. Features from $offset$ and $\phi$ provide global information between anchor vectors and patch embeddings. This is where the method gains positional/structural awareness, turning scalar importances + weighted embeddings into explicit geometric interactions usable by the transformer.

"Algorithm 1" is robust to top-K weight smoothing, indicating the anchor pathway encodes a distributed signal rather than relying on a few peak weights. In contrast, randomization/permutation of $s_j, b_j$ significantly degrades accuracy, confirming specificity of the learned importance.

## 3.3 POWER GATE METHOD WITH RESIDUAL MIXING

We explored inducing global quasi-positional information prior using pre-trained ViTs, rather than adding additional trainable params ( expect task-specific logit layer ) in this method.

One simple , yet powerful way to think about global information priors is entropy on tokens; a lower entropy of a token, should intuitively yield to amplified embedding value of token as uncertainty is low and embeddings to shrinks when entropy is higher. This behaviour is achieved by flipping entropy values , called confidence $c_j = 1 - H_j$; sign of embedding values were also accounted for in our algorithm.

---

**Algorithm 2** PFIM: Power Gate Modulation Method

---

1: Given $X \in \mathbb{N}xD$, compute aggregate vectors, $vector\_values \in \mathbb{N}$,containing raw score per token.
2: Compute Entropy Score of Tokens:
    a) $H_j = [- \sum Pr(x_j) * \log(Pr(x_j))] \div \log(D)$
    b) Flip the sign for confidence: $c_j = 1 - H_j$
3: Compute Sign-Aware X Power
    a) Bind $X->(0,1)$, so that $pow > 1$ always shrinks: $x_b = \tanh(x_j/\tau)$.
    b) Map exponent, $low\_score = alpha > 1$, $high\_score = alpha < 1$: $\alpha = \alpha\_low + (\alpha\_high - \alpha\_lo) * (1 - c_j)$.
    c) Sign-Safe Power: $x\_pow = sign(x\_b) * (abs(x\_b) + \epsilon)^\alpha$.
4: Residual Mix: $x\_mix = (1 - mix\%) * x_j + mix\% * x\_pow$

---

Interestingly, "Algorithm 2" is robust to top-K weight smoothing and inverse effects on embedding scaling (amplifying high-entropy tokens, damping low-entropy ones); where $H_j$ was directly used as $c_j$ instead of $c_j = 1 - H_j$, we report this as a deliberate control called $PFIM^{inv}$ that probes directionality. $PFIM^{inv}$ still outperformed vanilla ViT and was close to Single-PEG in both ablations and full training.

Likely because: (i) Any structured pre-MHSA token reweighting reshapes the token geometry fed to $QK^T$ ,yielding useful regularization. (ii) LayerNorm ensures uninformative scales are aptly absorbed, preventing collapse.

# 4 EXPERIMENTAL SETUP

## 4.1 DATASETS

CIFAR 10, 100 data sets from Krizhevsky (2009) were used for all ablations and main experiments. CIFAR's train sets were split to $70 - 30$ as train and validation for main experiment, and cifar test set was taken as-is. For ablations, we used 50% **stratified sample** of main experiment's training dataset: $train\_ablations$ and split validation of main experiment into $stratified$ $60 - 40$ as: $val\_ablations$ and $test\_ablations$. All experiments and ablations were run with pre-computed tensor data sets with Random Affine, Horizontal Flip and Normalization with data set mean and std.

## 4.2 ARCHITECTURE

Figure 1 shows RADAR block which produces single scale and shift vectors per token in $\in \mathbb{D}$, $CLS$ was not modulated and $\alpha$, $\beta$ are trainable parameters which start at 0.1. $APE$ injection post modulation of patch embeddings worked best, than before. Similarly, in PFIM ( Figure 2) patch embeddings are modulated with residual mixing followed by $APE$ injection. PFIM enures patch embeddings $x_j$ are bounded and power operation is sign-aware; it contains no trainable params, in computing $x_j^{pow}$.

## 4.3 BASELINES

$google/vitbasebasepatch16224$ checkpoint from Wu et al. (2020) was used for Vanilla ViT baseline, here as for Single-PEG , we used a $27x27$ kernel and trained it from scratch with a single convolution layer in PEG.

Table 1: Accuracy Metric Results: CIFAR10

| MODEL_TYPE | Num_Seeds | Hyper Parameters | Test_Mean_STD |
|---|---|---|---|
| Vanilla ViT | 4 | NA | $93.440 \pm 0.0006$ |
| Single PEG | 4 | K = 27 | $95.276 \pm 0.00063$ |
| RADAR SSA V1 | 4 | L2Norm, Soft max; Weighted Sum | $\mathbf{97.472} \pm 0.00058$ |
| PFIM | 4 | Entropy; Power Gate; mix% of 0.3 | $\mathbf{95.001} \pm 0.00085$ |
| PFIM | 2 | L2 Norm; Soft max | $\mathbf{94.73} \pm 9.5e - 05$ |
| $PFIM^{INV}$ | 3 | Entropy; Power Gate; Rev | $\mathbf{94.80} \pm 0.00064$ |

Table 2: Accuracy Metric Results: CIFAR100

| MODEL_TYPE | Num_Seeds | Hyper Parameters | Top1_Mean_Std | Top5_Mean_Std |
|---|---|---|---|---|
| Vanilla ViT | 4 | NA | $79.01 \pm 0.00232$ | $94.26 \pm 0.00059$ |
| Single PEG | 4 | K = 27 | $\mathbf{82.38} \pm 0.00223$ | $\mathbf{96.02} \pm 0.000818$ |
| RADAR SSA V1 | 4 | L2Norm, Soft max; Weighted Sum | $\mathbf{86.53} \pm 0.0035$ | $\mathbf{97.63} \pm 0.0017$ |
| PFIM | 4 | Entropy, Power Gate; mix% of 0.3 | $81.074 \pm 0.0014$ | $\mathbf{95.30} \pm 0.00080$ |

## 4.4 SETUP

For all runs(main or ablation), we used AdamW optimizer, Cosine LR scheduler with warm-up and chose best model based on smoothed validation set loss. In main experiment we employed early stopping with patience of 10 epochs , based on smoothed validation loss ( average of last 3 validation losses ), in ablations we used smaller patience ( 5) and fewer epochs 60 ( 20% of main ).

A single A100 GPU with capacity of 360 TFLOPs was used for all of our main experiments. For ablations, PFIM methods were able to run on single L4 GPU.

## 5 RESULTS

Single-PEG converges quickly ( by 26th epoch ) and stops training earlier than RADAR methods (stops around 31st epoch), as shown in Figure 3.Similarly, PFIM method also trains for slightly longer time than Single-PEG and early stops around 40th epoch, as shown in Figure 4.

Statistically significance was achieved at 95% confidence interval comparing each of our methods with baselines: Single-PEG and Vanilla ViT in Table 3. For TOST equivalence test, we used a small delta. A simple PFIM operation is found to be statistically equivalent to Single-PEG within a small $\delta \pm 0.01$; inspite of having $\mathbf{98}\%$ fewer training parameters in comparison.

While PFIM's Top1 performance is statistically equivalent to Single-PEG on CIFAR10, it suffers slightly on CIFAR100's Top1 but attains equivalence on Top5 ( Table 2 ). This shows that PFIM is robust across data sets and achieves on par performance even with simple linear injection of patch features.

## 6 ABLATION STUDY

Datasets for ablation study was chosen as explained in "Experimental Setup" section. Ablation study was conducted in a methodical three-stage process to systematically evaluate contributions of various components of the model.

### 6.0.1 STAGE 1: FOUNDATIONAL ARCHITECTURE COMPARISON

This stage aimed to identify the optimal ensemble configuration of custom models without APE. We compared the performance of various model combinations against a canonical, single-patch embedding generator (PEG) baseline to establish a performance benchmark, Table 5.

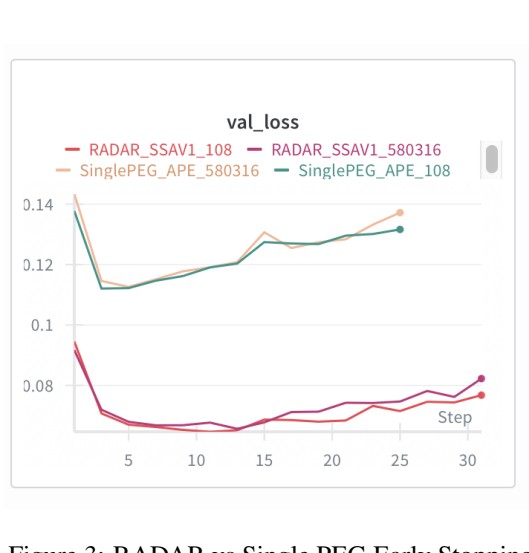

Figure 3: RADAR vs Single PEG Early Stopping

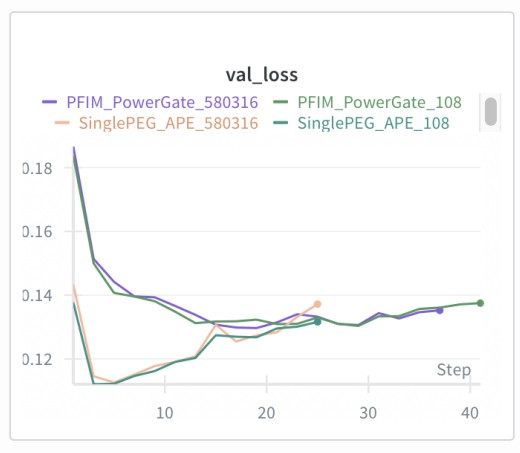

Figure 4: PFIM vs Single PEG Early Stopping

Table 3: Significance Results: CIFAR10

| Competitor_vs_Baseline | Num_Seeds | Our_Mean_SD | Baseline_Mean_SD | Test_Type | p-value |
|---|---|---|---|---|---|
| $[RADARSSA]\_vs\_[VanillaViT]$ | 4 | $97.4 \pm 0.0010$ | $93.4 \pm 0.00064$ | Paired T-Test. NH: $VVit >= RADAR$ | 1.411e-05 |
| $[RADARSSA]\_vs\_[SinglePEG]$ | 4 | $97.4 \pm 0.0010$ | $95.3 \pm 0.00072$ | Paired T-Test. NH: $SPEG >= RADAR$ | 5.87e-05 |
| $[PFIM]\_vs\_[VanillaViT]$ | 4 | $95.001 \pm 0.00085$ | $93.4 \pm 0.00064$ | Paired T-Test. NH: $VViT >= PFIM$ | 0.000169 |
| $[PFIM\_vs\_[SinglePEG]$ | 4 | $95.001 \pm 0.00085$ | $95.3 \pm 0.00072$ | Non-Inferiority Paired T-Test. NH: $\mu\_d <= -\delta$ | 0.00142 |
| $[PFIM]\_vs\_[SinglePEG]$ | 4 | $95.001 \pm 0.00085$ | $95.3 \pm 0.00072$ | TOST Paired Equivalence Test. NH: $\mu_d \leq -\delta \ \lor \ \mu_d \geq \delta$. | 0.00142 |

Table 4: Significance Results: CIFAR100

| Competitor_vs_Baseline | Num_Seeds | Our_Mean_SD | Baseline_Mean_SD | Test_Type | p-value |
|---|---|---|---|---|---|
| $[RADARSSA\_Top1]\_vs\_[VanillaViT\_Top1]$ | 4 | $86.53 \pm 0.0035$ | $79.013 \pm 0.0023$ | Paired T-Test. NH: $VVit >= RADAR$ | 1.847e-06 |
| $[RADARSSA\_Top1]\_vs\_[SinglePEG\_Top1]$ | 4 | $86.53 \pm 0.0035$ | $82.38 \pm 0.00223$ | Paired T-Test. NH: $SPEG >= RADAR$ | 0.00017 |
| $[PFIM\_Top1]\_vs\_[VanillaViT\_Top1]$ | 4 | $81.07 \pm 0.00146$ | $79.013 \pm 0.0023$ | Paired T-Test. NH: $VViT >= PFIM$ | 1.55e-05 |
| $[PFIM\_Top5]\_vs\_[SinglePEG\_Top5]$ | 4 | $95.30 \pm 0.00083$ | $96.02 \pm 0.00081$ | Non-Inferiority Paired T-Test. NH: $\mu\_d <= -\delta$ | 0.0265 |
| $[PFIM\_Top5]\_vs\_[SinglePEG\_Top5]$ | 4 | $95.30 \pm 0.00083$ | $96.02 \pm 0.00081$ | TOST Paired Equivalence Test. NH: $\mu_d \leq -\delta \ \lor \ \mu_d \geq \delta$. | 0.0265 |

Interestingly, Entropy and L2-Norm $->$ Soft max over patch tokens yielded similar performance in RADAR-SSA variant, with former being slightly better. When $APE$ is turned off Single-PEG emerges as strongest model, closely followed by RADAR Single Soft Anchor on CIFAR10 ( from Table 5). Since LOOSA and SSA had comparable performance, SSA variant was chosen as best performing variant in RADAR models as it uses fewer flops and has lower runtime complexity.

### 6.0.2 STAGE 2: ABSOLUTE POSITIONAL EMBEDDING (APE) INTEGRATION ANALYSIS

From best combination found in stage 1, we explore how model's performance changes when APE is injected, including single-PEG.

Similar to stage 1 ablations ( from Table 5 ) , SSA performs slightly better than LOOSA ; while significantly outperforming PFIM and Single-PEG when pre-trained $APE$ is injected ( from Table 6 ) on CIFAR10. Single-PEG benefits slightly from APE whereas our variants benefit drastically, proving that our methods are complimentary, but not replacement, to APE and they're better with positional information. $APE$ was turned on for main experiment for apples-to-apples comparison.

### 6.0.3 STAGE 3: COMPONENT ROBUSTNESS ANALYSIS

Stage 3 of our ablation work finds out robustness of our variants like Top-K weight smoothing, necessity of signals etc.

Both RADAR and PFIM are robust to alignment-shuffle ablations (random permutation of token features or importance scores), which leaves accuracy at baseline due to pre-norm LayerNorm canceling uncorrelated noise ( Table 7 ). However, turning off priors only at inference reveals their necessity: RADAR drops by **47.3** pp (82.73 $\rightarrow$ 35.43) and PFIM by **0.95** pp (59.83 $\rightarrow$ 58.88), confirming that gains arise specifically when priors remain content-aligned, as they reshape token geometry fed into $QK^T$ and attention weights.

Our modules are helpful when aligned and harmless when not: (i) necessity ablations confirm contribution (as accuracy falls when removed); (ii) alignment-shuffle show that uninformative priors are normalized away. Together with main results beating strong baselines, this demonstrates that RADAR and PFIM inject meaningful, content-aware signal rather than exploiting capacity or regularization artifacts. Similar ablations on CIFAR100 were tested and included in appendix.

## 7 CONCLUSION

We presented two lightweight, content-aware patch modulation modules for ViTs: RADAR (anchor-based distance priors) and PFIM (parameter-free importance scaling). Both operate pre-MHSA, keep the ViT backbone frozen, and add negligible overhead while reallocating attention toward semantically relevant regions.

On CIFAR-100, RADAR improves Top-1 accuracy by +7.5 pp and Top-5 by +3.3 pp over vanilla ViT, and by +4.1 pp / +1.6 pp over Single-PEG, while using **56%** fewer parameters. PFIM achieves +2.0 pp (Top-1) and +1.1 pp (Top-5) gains over vanilla ViT with **88%** fewer parameters than Single-

Table 5: Stage 1 Ablations Summary: CIFAR10

| MODEL_TYPE | APE_ON | Num_Seeds | Agg-Type | Seq_Select_Type | Test_Acc_Stats |
|---|---|---|---|---|---|
| RADAR Single Soft Anchor V1 | NO | 2 | L2Norm, Soft max | Weighted Sum | $82.73 \pm 0.00058$ |
| RADAR Leave-One-Out Soft Anchor | NO | 2 | L2Norm, Soft max | LOO Weighted Sum | $82.52 \pm 0.00198$ |
| RADAR SSA V1 | NO | 2 | Entropy | Weighted Sum | $81.72 \pm 0.00290$ |
| RADAR LOO-SA | NO | 1 | Entropy | LOO Weighted Sum | 82.052 |
| PFIM | NO | 2 | Entropy | Power Gate | $59.83 \pm 0.00146$ |
| PFIM | NO | 1 | L2Norm, Soft max | Power Gate | 61.812 |
| Single PEG (27x27) | NO | 2 | NA | NA | $91.36 \pm 0.00366$ |

Table 6: Stage 2 Ablations Summary: CIFAR10

| MODEL_TYPE | APE_ON | Num_Seeds | Agg-Type | Seq_Select_Type | Test_Accuracy |
|---|---|---|---|---|---|
| RADAR SSA V1 | YES | 1 | L2 Norm,Soft max | Weighted Sum | **97.775** |
| RADAR LOO SA | YES | 1 | L2 Norm, Soft max | LOO Weighted Sum | 97.405 |
| PFIM | YES | 1 | Entropy | Power Gate | **95.275** |
| PFIM | YES | 1 | L2 Norm; Soft max | Power Gate | **95.291** |
| Single PEG ( 27 x 27 ) | YES | 1 | NA | NA | 95.804 |

Table 7: Stage 3 Ablations Summary: CIFAR 10

| MODEL_TYPE | APE_ON | Num_Seeds | Agg-Type | Seq_Select_Type | Ablation_Type | Params | Test_Acc_Stats |
|---|---|---|---|---|---|---|---|
| RADAR SSA V1 | NO | 2 | L2Norm, Soft max | Weighted Sum | TopK Weight Smoothing | Alpha = (0.8, 1.0); TopK Sequences = 98 | $83.127 \pm 0.0017$ |
| RADAR SSA V1 | NO | 2 | L2Norm, Soft max | Weighted Sum | Necessity Test | Sj, Bj both set to 0 | $35.43 \pm 0.00170$ |
| RADAR SSA V1 | NO | 1 | L2Norm, Soft max | Weighted Sum | Random Permutation of Sj,Bj | NA | 82.07 |
| PFIM | NO | 2 | Entropy | Power Gate | TopK Weight Smoothing; Alpha = (0.8, 1.0) | TopK Sequences = 98 | $60.56 \pm 0.00190$ |
| PFIM | NO | 2 | Entropy | Power Gate | Necessity Test | mix% = 0 | $58.88 \pm 0.00217$ |

PEG, matching its performance within a small delta(0.01). Additionally, on CIFAR-10 both modules deliver consistent, statistically significant gains, confirming robustness across datasets.

While our methods show strong improvements under absolute positional encoding setting, Single-PEG performs better when APE is turned off.Possibly because PEG learns transformations on patch embeddings , while we learn distance-based features and only provide complementary modulations to patch embeddings rather than projecting them directly. Nevertheless, the results demonstrate that RADAR and PFIM provide parameter-efficient, low-budget improvements that outperform strong baselines.

# 8 FUTURE WORK

We will extend the distance signal in three directions. First, we will add 2D patch coordinates to features and test them with and without absolute positional encodings (APE), and we will check whether coordinate-anchor distances give extra gains beyond patch–patch distances. Second, we will replace a single anchor with a small grid of anchors to study grid–grid distance modulation. Third, we will vary where we inject these signals (alternating versus sequential transformer blocks) and examine the effect on attention maps, stability, and accuracy. In parallel, we will probe CNN backbones by inserting the same patch-neighborhood priors at different depths to see which layers benefit most. Finally, we will validate on larger datasets and explore compatibility with relative positional schemes such as iRPE and RoPE, measuring accuracy–efficiency trade-offs and when distance cues and relative positions are complementary versus redundant.

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

Table 8: Ablations Summary: CIFAR 100

| MODEL_TYPE | APE_ON | Num_Seeds | Ablation_Type | Params | Top1_Stats | Top5_Stats |
|---|---|---|---|---|---|---|
| RADAR SSA V1 | NO | 1 | Necessity Test | $s_j, b_j = 0, 0$ | 19.95 | 44.75 |
| RADAR SSA V1 | NO | 1 | Align Test | Random Permutation of Sj,Bj | 60.167 | 86.226 |
| RADAR SSA V1 | YES | 1 | Align Test | Random Permutation of Sj,Bj | 88.379 | 98.567 |
| RADAR LOOSA | NO | 1 | Align Test | Random Permutation of Sj,Bj | 60.321 | 87.138 |
| RADAR LOOSA | YES | 1 | Align Test | Random Permutation of Sj,Bj | 88.292 | 98.477 |
| PFIM | NO | 1 | Align Test | Random Permutation of importance weights | 35.027 | 63.703 |
| PFIM | YES | 1 | Align Test | Random Permutation of importance weights | 82.543 | 96.544 |

Table 9: FLOPs vs Params By Model

| MODEL_TYPE | Dataset | Train_FLOPS | Num_Train_Params | Train_Params_MB |
|---|---|---|---|---|
| RADAR SSA V1 | CIFAR10 | 55,208,252,064 | **207,116** | 0.7900848388671875 |
| RADAR SSA V1 | CIFAR100 | 55,208,597,664 | 276,326 | 1.0541000366210938 |
| PFIM | CIFAR10 | 18,338,003,702 | **7,690** | 0.02933502197265625 |
| PFIM | CIFAR100 | 18,338,211,062 | 76,900 | 0.2933502197265625 |
| Single-PEG | CIFAR10 | 18,665,249,808 | **568,330** | 2.168006896972656 |
| Single-PEG | CIFAR100 | 18,665,457,168 | **637,540** | 2.4320220947265625 |

Hugo Touvron, Matthieu Cord, Matthijs Douze, Francisco Massa, Alexandre Sablayrolles, and Hervé Jégou. Training data-efficient image transformers & distillation through attention. In *International Conference on Machine Learning (ICML)*, volume 139 of *Proceedings of Machine Learning Research*, pp. 10347–10357, 2021a. URL https://proceedings.mlr.press/v139/touvron21a.html.

Hugo Touvron, Matthieu Cord, Alexandre Sablayrolles, Gabriel Synnaeve, and Hervé Jégou. Going deeper with image transformers. In *ICCV*, pp. 12164–12174, 2021b. URL https://openaccess.thecvf.com/content/ICCV2021/papers/Touvron_Going_Deeper_With_Image_Transformers_ICCV_2021_paper.pdf.

Bichen Wu, Chenfeng Xu, Xiaoliang Dai, Alvin Wan, Peizhao Zhang, Zhicheng Yan, Masayoshi Tomizuka, Joseph Gonzalez, Kurt Keutzer, and Peter Vajda. Visual transformers: Token-based image representation and processing for computer vision, 2020.

Yunyang Xiong, Zhanpeng Zeng, Rudrasis Chakraborty, Mingxing Tan, Glenn Fung, Yin Li, and Vikas Singh. Nyströmformer: A nyström-based algorithm for approximating self-attention. In *AAAI*, pp. 14138–14148, 2021. URL https://arxiv.org/abs/2102.03902.

Ling Zhu, Zhenhua Ma, Jie Wang, Jingdong Zhang, Biao Wang, Shifeng Tang, Yansong Zhang, and Dacheng Tao. Biformer: Vision transformer with bi-level routing attention. In *CVPR*, pp. 10321–10331, 2023. URL https://openaccess.thecvf.com/content/CVPR2023/html/Zhu_BiFormer_Vision_Transformer_With_Bi-Level_Routing_Attention_CVPR_2023_paper.html.

## A  APPENDIX

Figure 5 shows the distribution of FLOPs per sample for each of custom models excluding Vanilla ViT. RADAR-SSA needs more FLOPs ( $+67\%$ to be precise ) than Single-PEG while PFIM uses fewer FLOPs ( 2% ) than Single-PEG with comparable performance ( from Table 9 ).

Figure 6 shows the distribution of trainable params needed for each model excluding Vanilla ViT. It is interesting to note that even though single PEG ( k = 27 ) uses depth-wise kernels, it still uses significantly more parameters ( $+64\%$ ) and memory than RADAR-SSA with lower test time performance. Where as PFIM uses 98% fewer params with comparable performance to Single-PEG within a small $\delta \pm 0.01$.

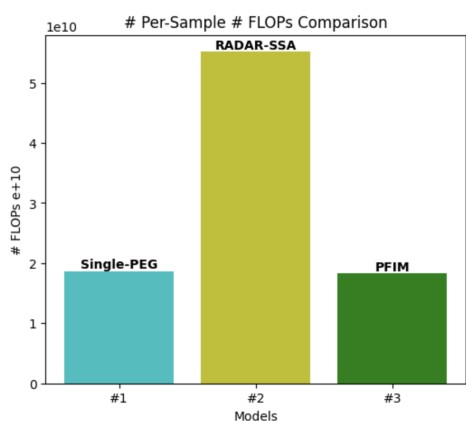

Figure 5: Number of FLOPs per Sample, for all custom models. CIFAR10

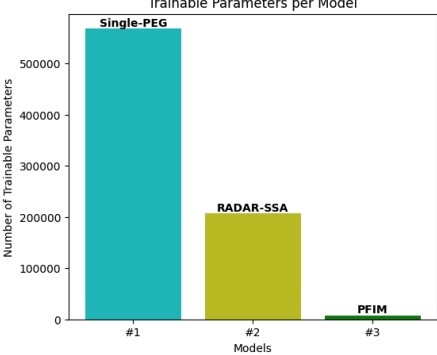

Figure 6: Number of Trainable params, for all custom models

## A.1 TRAINING TIME ROBUSTNESS

In addition to random permutation and necessity ablation tests, we also conducted training-time ablation tests such as top-K importance weight smoothing ( from Table 7 ) i.e., top-K importance weights,which contribute to creating $s_j, b_j$ in RADAR and importance weights in PFIM, were smoothed both during training and inference, this did not effect model performance on test ablation set.

These observations are in-line with our hypothesis that when injecting such modulations pre-encoder, first LayerNorm largely absorbs any affine transformations, making it robust to such shifts, irrespective of type of modulation.

## A.2 CODE BASE AND TIPS TO REPRODUCE

All hyper parameters and epoch details come from config yaml file ( configs_ablations.yaml and configs_train.yaml ) and two separate .py files exist with ViT code, one for ablations and one for main experiment ( ViT_ablations.py and Custom_VIT.py ).

These files are orchestrated from another set of .py files specific to ablations and main experiment ( run_benchmarks_ablations.py and run_benchmarks.py). We strongly recommend pre-computing tensors before training with all transformations and saving them in respective folders, with specific seeds ( 108 ) for exact reproducibility.

Suggested flow of operations are: Pre-Compute Tensors from specific seeds $->$ Choose the Config to run from configs file ( pfim_normal, pfim_necessity etc ) $->$ run respective benchmark file, with name from configs yaml. Users can choose to log metrics and model to Wandb optionally ( log_metrics = True, log_model = True ) in run_benchmarks file.

## A.3 LLM USAGE

We employed LLMs strictly as **productivity aids**—to generate code skeletons, resolve minor bugs, and paraphrase or shorten text for readability. Importantly, LLMs were **not** involved in the ideation or methodological design stages. The conception and development of our two techniques, RADAR and PFIM, are **entirely** our **own original** contributions.

