# OpenReview forum: "RADAR and PFIM: Content-Prior Patch-Space Modulation for Efficient Vision Transformers"
_ICLR.cc/2026/Conference — ICLR 2026 Conference Withdrawn Submission_

### Official Review · Reviewer_uj3p · 2025-10-26

**Soundness:** 2
**Presentation:** 1
**Contribution:** 2
**Rating:** 2
**Confidence:** 3

**Summary:**

This paper introduces two lightweight, content-aware patch modulation techniques: RADAR (Relational Anchor-Distance Attentional Re-weighting) and PFIM (Parameter-Free Importance Modulation), for enhancing the efficiency and performance of Vision Transformers (ViTs).
Both modules operate pre-MHSA (Multi-Head Self-Attention), injecting content-aligned priors computed directly from pretrained patch embeddings, without modifying the ViT backbone or adding significant parameters.

Experiments on CIFAR-10 and CIFAR-100 demonstrate that:

RADAR boosts Top-1 accuracy by +7.5 pp over vanilla ViT and +4.1 pp over Single-PEG with 56% fewer parameters.

PFIM achieves +2.0 pp Top-1 and +1.1 pp Top-5 improvements with up to 98% parameter reduction relative to Single-PEG.
Both approaches yield statistically significant gains under paired t-tests and equivalence tests across multiple random seeds.

**Strengths:**

1. novel concept: introduces content-prior modulation in patch space, distinct from standard positional encoding.
2. lightweight and modular: both RADAR and PFIM are drop-in additions requiring minimal computation or parameter cost.
3. Transparent reporting: FLOPs, parameter counts, and reproducibility instructions are explicitly detailed in the appendix.
4. Interpretable design: distance-based relational priors and entropy-based scaling have clear intuitive grounding.

**Weaknesses:**

1. limited dataset scope: only CIFAR-10/100 experiments are presented. Evaluating on larger datasets would confirm scalability.
2. Minor writing inconsistencies: some notation, spacing, and punctuation reduce readability.
3. Underexplored visualization: the paper could include attention heatmaps or token-importance visualizations to illustrate how modulation changes model behavior.
4. The teaser figure needs some improvement.

**Questions:**

1. Could PFIM be integrated into dynamic token pruning or distillation pipelines to further reduce inference cost or not?
2. How sensitive are both methods to $\alpha$, $\beta$ initialization and mix% hyperparameters?
3. Have you tested RADAR/PFIM on larger backbones (e.g., ViT-B/32) or datasets like ImageNet?

---

### Official Review · Reviewer_myTj · 2025-10-30

**Soundness:** 2
**Presentation:** 2
**Contribution:** 2
**Rating:** 2
**Confidence:** 3

**Summary:**

This paper proposes two lightweight pre-MHSA modules — **RADAR** and **PFIM** — designed to provide content-aware modulation for Vision Transformers (ViTs). RADAR introduces anchor-conditioned distance priors, while PFIM applies parameter-free importance scaling based on patch entropy. The methods target parameter-efficient gains on CIFAR-10/100 with frozen ViT backbones.

**Strengths:**

- Reasonable motivation for pre-attention content priors.
- Lightweight implementation with negligible parameter cost.
- Shows modest gains over vanilla ViT.

**Weaknesses:**

1. **Very limited experiments.**
   Only evaluated on CIFAR-10/100; no ImageNet or larger models. Reported gains are minor and dataset scale is insufficient to claim significance.

2. **Poor figure and presentation quality.**
   Figures are unclear and inconsistent; architecture diagrams fail to illustrate the design clearly.

3. **Unconvincing ablation and analysis.**
   Statistical tests and significance claims are weak; experiments are run on too few seeds.

4. **Missing comparisons.**
   Absent evaluation against contemporary efficient adaptation methods (LoRA, AdaptFormer, VPT, etc.).

5. **Weak writing quality.**
   Numerous grammatical issues, inconsistent structure, and overcomplicated pseudo-math.

**Questions:**

Please refer to the Weaknesses

---

### Official Review · Reviewer_9X5f · 2025-10-30

**Soundness:** 2
**Presentation:** 1
**Contribution:** 2
**Rating:** 2
**Confidence:** 4

**Summary:**

This work introduces RADAR and PFIM, two lightweight methods designed to bias a transformer's attention toward task-relevant tokens. Both methods achieve this by modifying patch embeddings based on global context before they are processed by the transformer layers.

RADAR uses pretrained patch embeddings to compute "anchor tokens" and then uses the resulting anchor distance offsets to modulate each patch's features. PFIM, in contrast, scales patch embeddings based on an "importance" score, which is computed using token-wise, task-specific entropy. The authors show that both RADAR and PFIM outperform Vanilla ViTs and Single PEG on CIFAR-10 and CIFAR-100 classification.

**Strengths:**

* Simple and Practical: Both RADAR and PFIM are easy to add to existing ViTs, and do not require major architecture changes.
* Performance: Both methods show improved performance compared to vanilla ViT and Single-PEG on CIFAR classification

**Weaknesses:**

1. The experiments are confined to low-resolution (32x32) CIFAR datasets. To demonstrate practical relevance and scalability, the authors should benchmark RADAR and PFIM on ImageNet, which is the standard for vision classification models. It also makes less sense to benchmark on CIFAR when using tokens from models pretrained on ImageNet.
2. The ablation studies are limited, as they only test specific hyperparameter combinations. A more rigorous approach would involve systematic hyperparameter sweeps (varying one while holding others fixed) to isolate the impact of each. Furthermore, critical experimental details are omitted, including learning rate, batch size, and cosine scheduling parameters.
3. If pretrained tokens are already required, then the authors should use direct feeding of these tokens (with regular PEs) as a baseline. The pretrained model's performance (instead of just Vanilla ViT) should also be measured.
4. All experiments are conducted only on classification tasks. Can the method be applied to object detection, and semantic segmentation?
5. Both methods depend on pretrained representations, yet they are only tested with a single pretrained model. Can the authors test robustness using other pretrained models (e.g., DINO), which might offer more semantically rich token representations?

**6. Poor Clarity and Organization**

The paper's organization and writing are unclear, making it extremely difficult to follow.

**Figures**: Figures 1 and 2 are small, disorganized, and lack informative captions. They are not referenced until late in the paper (Section 4.2) and waste space depicting the standard transformer architecture instead of clearly illustrating the novel RADAR and PFIM components.

**Notation**: The mathematical notation is poor. It is inconsistent (e.g., $s$_$j$ on line 141 vs. $s_j$ on line 056; $X_j$ in Algorithm 1 vs. $x_j$ elsewhere) and unclear, with many variables introduced but never defined (e.g., $x_j, \alpha, \beta, s_j, b_j$ on line 056).

**Missing Context and Citations**: Key related methods (e.g., FiLM) are poorly explained, while many others (T5, ALiBi, RoPE, Swin, LoRA) are mentioned without references. Furthermore, numerous acronyms (PEG, SSA, LOOSA, APE) are used without ever being introduced.

Organization and clarity are sufficient reasons for rejection for me; the paper is not in a state to be a published as-is.

**Questions:**

1. How are vector_values obtained in Algorithm 1? Are they also used in Algorithm 2 or is this a typo?
2. It is not clear to me what the 'entropy' score of a token is for PFIM. If $X_j$ are tokens, then how is $\text{Pr}(X_j)$ computed, and what does it represent? It also not clear what is being summed in the expression for $H_j$. I'm skeptical this would be a good measure of importance, as opposed to other saliency measures. It would be nice if the authors could try using aggregated attention scores (e.g. average of CLS to patches, across layers/heads) from powerful SSL models such as DINO.

Please also refer to the weaknesses.

---

### Official Review · Reviewer_6vGn · 2025-10-31

**Soundness:** 2
**Presentation:** 1
**Contribution:** 1
**Rating:** 2
**Confidence:** 4

**Summary:**

This paper proposes two modules (RADAR and PFIM) that adapt a frozen ViT by injecting “content-aware quasi-positional priors” into patch embeddings before MHSA, using anchor-based distance features or entropy-based scaling. Experiments are conducted on CIFAR-10/100 using ViT-Base pretrained on ImageNet.

**Strengths:**

- The paper introduces a parameter-efficient ViT adaptation approach, which is shown to work on CIFAR10 and CIFAR 100.
- The proposed methods seem reasonable.

**Weaknesses:**

- Weak empirical evaluation: The experiments are limited to CIFAR-10/100, which are small and outdated for evaluating ViTs. The backbone used is ViT-Base (86M parameters) fine-tuned on CIFAR—a setting that is not meaningful, as small CNNs or ViT-Tiny models trained from scratch can achieve comparable performance. Despite broad claims, no additional benchmarks are included. Moreover, the paper positions itself as a parameter-efficient adaptation method but omits standard PEFT baselines such as LoRA, VPT, AdaptFormer, adapters. The only comparison is with Single-PEG, which is insufficient.

- Misleading compute claims: RADAR increases FLOPs by 67% compared to the baseline and requires more epochs to converge, contradicting the claim of “negligible overhead.” PFIM’s efficiency gains do not justify the combined claims for both modules.

- Limited novelty and conceptual clarity: The proposed method closely resembles existing techniques, such as ConViT, content-aware positional embeddings, and token-importance weighting. PFIM’s entropy scaling appears ad-hoc, and the paper’s claims of novelty are overstated.

**Questions:**

Could the authors evaluate robustness to different pretrained checkpoints, patch sizes, training hyperparameters, and model architectures? This would help confirm that the method is not tuned to a single CIFAR training recipe.

---

### Note · Authors · 2025-11-13

I have read and agree with the venue's withdrawal policy on behalf of myself and my co-authors.